# Bipartite Graph Pre-training for Unsupervised Extractive Summarization with Graph Convolutional Auto-Encoders

**Qianren Mao[1], Shaobo Zhao[2], Jiarui Li[2], Xiaolei Gu[2], Shizhu He[3], Bo Li[1,4], Jianxin Li[1,4,*]**

[1] Zhongguancun Laboratory, Beijing, P.R.China.
[2] School of Software, Beihang University, Beijing, P.R.China.
[3] Institute of Automation, Chinese Academy of Sciences, Beijing, P.R.China.
[4] School of Computer Science and Engineering, Beihang University, Beijing, P.R.China.
{maoqr,libo,lijx}@zgclab.edu.cn, shizhu.he@nlpr.ia.ac.cn
{zsb18377239,ljr19231244,gxl19373188}@buaa.edu.cn

## Abstract

Pre-trained sentence representations are crucial for identifying significant sentences in unsupervised document extractive summarization. However, the traditional two-step paradigm of *pre-training* and *sentence-ranking*, creates a gap due to differing optimization objectives. To address this issue, we argue that utilizing pre-trained embeddings derived from a process specifically designed to optimize cohesive and distinctive sentence representations helps rank significant sentences. To do so, we propose a novel graph pre-training auto-encoder to obtain sentence embeddings by explicitly modelling intra-sentential distinctive features and inter-sentential cohesive features through sentence-word bipartite graphs. These pre-trained sentence representations are then utilized in a graph-based ranking algorithm for unsupervised summarization. Our method produces predominant performance for unsupervised summarization frameworks by providing summary-worthy sentence representations. It surpasses heavy BERT- or RoBERTa-based sentence representations in downstream tasks.

## 1 Introduction

Unsupervised document summarization involves generating a shorter version of a document while preserving its essential content (Nenkova and McKeown, 2011). It typically involves two steps: pre-training to learn sentence representations and sentence ranking using sentence embeddings to select the most relevant sentences within a document.

Most research focuses on graph-based sentence ranking methods, such as TextRank (Mihalcea and Tarau, 2004) and LexRank (Erkan and Radev, 2004), to identify the significant sentence by utilizing topological relations. Continual improvements have been demonstrated by several attempts (Narayan et al., 2018; Zhou et al., 2018; Wang et al., 2019; Xiao and Carenini, 2019; Wang

---
[*] Jianxin Li is the corresponding author.

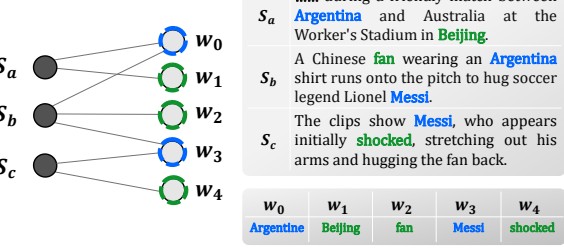

Figure 1: The graph structure of bipartite sentence-word graph. The sentences connect with unique nodes (monopolized by a single sentence node), and common nodes (shared by multiple sentence nodes).

et al., 2020), modelling graph-based ranking methods through a global view of the document.

For unsupervised document summarization, learning semantic sentence embeddings is crucial, alongside the sentence ranking paradigm. Textual pre-training models like skip-thought model (Kiros et al., 2015), TF-IDF, and BERT (Devlin et al., 2019) generate sentential embeddings, enabling extractive systems to produce summaries that capture the document's central meaning (Yasunaga et al., 2017; Xu et al., 2019; Jia et al., 2020; Wang et al., 2020). By combining sentence representations generated from pre-trained language models, prominent performances have been achieved with graph-based sentence ranking methods (Zheng and Lapata, 2019; Liang et al., 2021; Liu et al., 2021).

Despite the effectiveness of graph-based ranking methods that incorporate pre-trained sentential embeddings, there are some underexplored issues. Firstly, a significant gap exists between the two-step paradigm of *textual pre-training* and *sentence graph-ranking*, as the optimization objectives diverge in these two steps. The pre-trained framework is primarily designed to represent sentences with universal embeddings rather than summary-worthy features. By relying solely on the universal embeddings, the nuanced contextual information of the document may be overlooked, resulting in sub-

optimal summaries. Secondly, the existing graph formulation (e.g., GCNs (Bruna et al., 2014)) only encodes distinctive sentences but not necessarily cohensive ones, which may limit the extraction of summary-worthy sentences.

In summarization, cohensive sentences reveal how much the summary represents a document, and distinctive sentences involve how much complementary information should be included in a summary. To exemplify how these sentence features come from words, we analyze a sentence-word bipartite graph as depicted in Figure 1.

- The connections $S_a - w_1, S_b - w_2, S_c - w_4$ capture **intra**-sentential information, where the unique word nodes $w_1 = Bejing$, $w_2 = fan$, $w_4 = shocked$ contribute distinctive features to their respective sentence nodes $S_a, S_b, S_c$.

- The connections $S_a - w_0, S_b - w_0, S_b - w_3, S_c - w_3$ capture **inter**-sentential information, where the shared word nodes $w_0 = Argentina$, $w_3 = Messi$ contains cohensive features for their connected sentence nodes $S_a, S_b, S_c$.

Clearly, a sentence's unique features come from its individual word nodes, while its cohensive features come from shared word nodes with other sentences. Based on this observation, we argue that optimizing cohensive and distinctive sentence representations during pre-training is ultimately beneficial for ranking significant sentences in downstream extractive summarization. To achieve this, we propose a novel graph pre-training paradigm using a sentence-word bipartite graph with graph convolutional auto-encoder (termed as Bi-GAE[1]) to learn sentential representations.

In detail, we pre-train the bipartite graph by predicting the word-sentence edge centrality score in self-supervision. Intuitively, more unique nodes imply smaller edge weights, as they are not shared with other nodes. Conversely, when there are more shared nodes, their edge weights tend to be greater. We present a novel method for bipartite graph encoding, involving the concatenation of an inter-sentential $GCN^{inter}$ and an intra-sentential $GCN^{intra}$. These two GCNs allocate two encoding channels for aggregating inter-sentential cohesive features and intra-sentential distinctive features during pre-training. Ultimately, the pre-trained sen-

tence node representations are utilized for downstream extractive summarization.

Our pre-trained sentence representations obtain superior performance in both single document summarization on the CNN/DailyMail dataset (Hermann et al., 2015) and multiple document summarization on the Multi-News dataset (Sandhaus, 2008) within salient extractive summarization frameworks. i) To our knowledge, we are the first to introduce the bipartite word-sentence graph pre-training method and pioneer bipartite graph pre-trained sentence representations in unsupervised extractive summarization. ii) Our pre-trained sentence representation excels in downstream tasks using the same summarization backbones, surpassing heavy BERT- or RoBERTa-based representations and highlighting its superior performance.

## 2 Background & Related Work

### 2.1 Sentence Ranking Summarization

Traditional extractive summarization methods are mostly unsupervised (Yin and Pei, 2015; Nallapati et al., 2017; Zheng and Lapata, 2019; Zhong et al., 2019; Mao et al., 2022). Among them, graph-based sentential ranking methods are widely used. Two popular algorithms for single-document summarization are unsupervised LexRank (Erkan and Radev, 2004) and TextRank (Mihalcea and Tarau, 2004), estimating the centrality score of each sentence node among the textual context nodes.

In contrast to LexRank and TextRank constructing an undirected sentence graph, the model of PACSUM (Zheng and Lapata, 2019) builds a directed graph. Its sentence centrality is computed by aggregating its incoming and outgoing edge weights:

$$Centrality(s_i) = \lambda_1 \sum_{j<i} e_{i,j} + \lambda_2 \sum_{j>i} e_{i,j}, \quad (1)$$

where hyper-parameters $\lambda_1$, $\lambda_2$ are different weights for forward- and backward-looking directed edges and $\lambda_1 + \lambda_2 = 1$. $e_{i,j}$ is the weights of the edges $e_{i,j} \in E$ and is computed using word co-occurrence statistics, such as the similarity score. Building upon the achievements of PACSUM(Zheng and Lapata, 2019), recent models such as FAR (Liang et al., 2021) and DASG (Liu et al., 2021) have aimed to improve extractive summarization by integrating centrality algorithms. These models primarily focus on seeking central sentences based on semantic facets (Liang et al., 2021) or sentence positions (Liu et al., 2021).

---

[1]Code and data available at: *https://github.com/ OpenSUM/BiGAE*.

## 2.2 Sentential Pre-training

PLM's pre-training, such as BERT and GPT, is crucial for identifying meaningful sentences in downstream summarization tasks. The previously mentioned graph-based summarization methods, such as PacSum(Zheng and Lapata, 2019), FAR(Liang et al., 2021), and DASG (Liu et al., 2021) utilize pre-trained BERT representations for sentence ranking. STAS (Xu et al., 2020) takes a different approach by pre-training a Transformer-based LM to estimate sentence importance. However, STAS is not plug-and-play and requires a separate pre-training model for each downstream task.

Despite the success of the aforementioned unsupervised extractive summarization methods, it still maintains a gap between the PLMs' pre-training and the downstream sentence ranking methods. Additionally, low-quality representations can result in incomplete or less informative summaries, negatively affecting their quality. Pre-training models typically produce generic semantic representations instead of generating summary-worthy representations, which can result in suboptimal performance in unsupervised summarization tasks.

## 3 Methodology

In what follows, we describe our pre-training model BI-GAE (as shorthand for Bipartite Graph Pre-training with Graph Convolutional Auto-Encoders ) used for unsupervised extractive summarization. We will introduce bipartite graph encoding and the pre-training procedure using our BI-GAE. Ultimately, we will utilize the pre-trained sentence representations for the downstream unsupervised summarization.

## 3.1 Document as a Bipartite Graph

Formally, We denote the constructed bipartite word-sentence graph $\mathcal{G} = \{\mathcal{V}, \mathcal{A}, \mathcal{E}, \mathbf{X}\}$, where $\mathcal{V} = \mathcal{V}_w \cup \mathcal{V}_s$. Here, $\mathcal{V}_w$ denotes $|\mathcal{V}_w| = n$ unique words of the document and $\mathcal{V}_s$ corresponds to the $|\mathcal{V}_s| = m$ sentences in the document. $\mathcal{A} = \left\{e_{11}, ..., e_{i,j}, ..., e_{nm}\right\}$ defines the adjacency relationships among nodes, and $e_{i,j} \in \{0, 1\}^{n \times m}$ indicates the edge weight from source node $i$ to target node $j$. $\mathbf{X} \in \mathbb{R}^{(n+m) \times d}$, is termed as a matrix containing the representation of all nodes. The node representations will be iteratively updated by aggregating summary-worthy features (intra-sentential and inter-sentential messages) between word and sentence nodes via the bipartite graph autoencoder.

## 3.2 Bipartite Graph Pre-training

We reform the original VGAE (Kipf and Welling, 2016) pre-training framework by optimizing edge weight prediction in bipartite graphs. The pre-training optimizer learns to fit the matrices between the input weighted adjacency matrix and the reconstructed adjacency matrix in a typical way of self-supervised learning. By integrating an intra-sentential $\text{GCN}^{intra}$ and an inter-sentential $\text{GCN}^{inter}$ in the VGAE (Kipf and Welling, 2016) self-supervised framework, our pre-training method enables effective aggregation of intra-sentential and inter-sentential information, allowing for the representation of high-level summary-worthy features in the bipartite graph pre-training.

**Bipartite Graph Initializers.** Let $\mathbf{X}_w \in \mathbb{R}^{(n) \times d_w}$ and $\mathbf{X}_s \in \mathbb{R}^{(m) \times d_s}$ represent the input feature matrix of the word and sentence nodes respectively, where $d_w$ and $d_s$ are the dimension of word embedding vector and sentence representation vector respectively. We first use convolutional neural networks (CNN) (LeCun et al., 1998) with different kernel sizes to capture the local n-gram feature for each sentence $S_i^C$ and then use the bidirectional long short-term memory (BiLSTM) (Hochreiter and Schmidhuber, 1997) layer to get the sentence-level feature $S_i^L$. The concatenation of the CNN local feature and the BiLSTM global feature is used as the sentence node initialized feature $\mathbf{X}_{S_i} = [S_i^C; S_i^L]$. The initialized representations are used as inputs to the graph autoencoder module.

**Bipartite Graph Encoder.** To model summary-worthy representations, we encode the bipartite graph by a concatenation of an intra-sentential $\text{GCN}^{intra}$ and an inter-sentential $\text{GCN}^{inter}$, in which two GCNs assign two encoding channels for aggregating intra-sentential distinctive features and inter-sentential cohesive features. The $\text{GCN}^{intra}$ ($\mathbf{H}^0 = \mathbf{X}, \mathbf{A}_{weight}, \mathbf{\Theta}$), can be seen as a form of message passing to aggregate intra-sentential distinctive features. The first $\text{GCN}^{intra}$ layer generates a lower-dimensional feature matrix. Its node-wise formulation is given by:

$$\mathbf{h}_j^{intra} = \mathbf{\Theta}^\top \sum_{i \in N(u) \cup \{j\}} \frac{1}{\sqrt{\tilde{d}_i \tilde{d}_j}} e_{i,j} \mathbf{h}_i^{intra}, \quad (2)$$

where $e_{i,j} \in \mathbf{A}_{weight}$ denotes the edge weight from source node $i$ to target node $j$. Here we use the betweenness centrality [2] as the edge

---

[2] https://networkx.org/documentation/latest/reference/algorithms/centrality.html

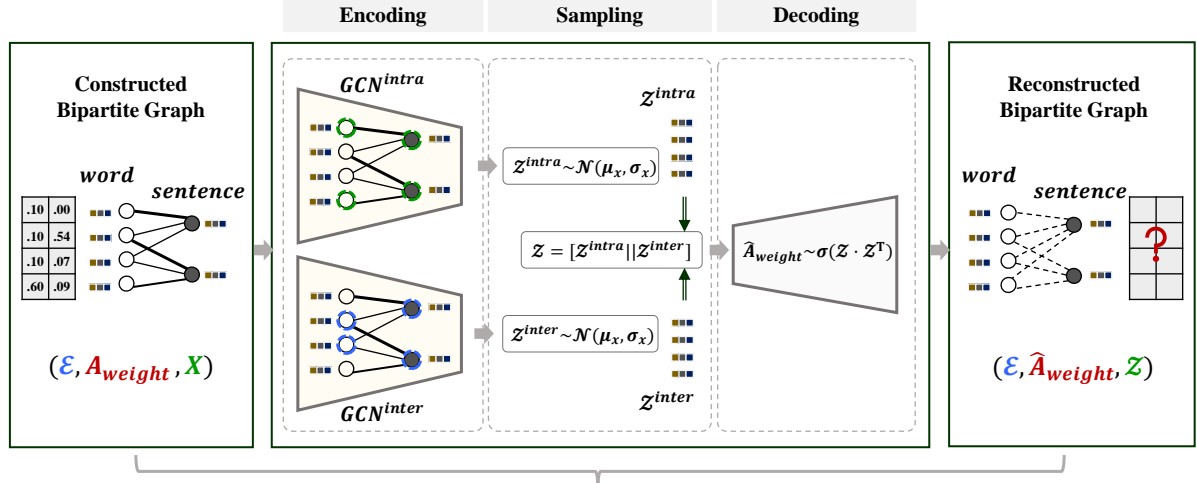

Figure 2: Overall architecture of our pre-training model Bɪ-GAE. We construct a sentence-word bipartite graph to optimize both distinctive intra-sentential and cohensive inter-sentential nodes, by predicting the word-sentence edge centrality scores using a self-supervised graph autoencoder.

weights. The betweenness centrality of an edge is the sum of fractions of the shortest paths passing through it. The first $GCN^{intra}$ layer makes features of neighbour nodes with fewer association relationships aggregated and enlarged and outputs a lower-dimensional feature matrix $\mathbf{H}$. Then, the second $GCN^{intra}$ layer generates $\mu^{intra} = GCN_\mu\left(\mathbf{H}^{intra}, \mathbf{A}_{weight}\right)$ and $log(\sigma^{intra})^2 = GCN_\sigma\left(\mathbf{H}^{intra}, \mathbf{A}_{weight}\right)$.

The $GCN^{inter}$ ($\mathbf{H}^0 = \mathbf{X}, \mathbf{A}_{weight}, \mathbf{\Theta}$) can be seen as a form of message passing to aggregate inter-sentential cohensive features:

$$\mathbf{h}_j^{inter} = \mathbf{\Theta}^\top \sum_{i \in N(u) \cup \{j\}} \frac{\sqrt{\tilde{d}_i}}{\sqrt{\tilde{d}_j}} e_{i,j} \mathbf{h}_i^{inter}. \quad (3)$$

The graph convolution operator $GCN^{inter}$ will aggregate neighbour node features with more association relationships aggregated and enlarged. Analogously, we can obtain $\mu^{inter}$ and $log(\sigma^{inter})^2$, which are parameterized by the two-layer $GCN^{inter}$.

Then we can generate the latent variable $\mathbf{Z}$ as output of bipartite graph encoder by sampling from $GCN^{inter}$ and $GCN^{intra}$ and then concatenating sampled two latent variables $\mathbf{Z}^{inter}$ and $\mathbf{Z}^{intra}$:

$$q(\mathbf{Z}^{inter} \| \mathbf{Z}^{intra}) = \prod_{i=1}^N q(\mathbf{z}_i^{inter}) \prod_{i=1}^N q(\mathbf{z}_i^{intra}), \quad (4)$$

where $q(\mathbf{z}_i^{inter})$ and $q(\mathbf{z}_i^{intra})$ are from two GCNs, satisfying independent distribution conditions. Here,

$$q(\mathbf{z}_i^{inter}) = \mathcal{N}\left(\mathbf{z}_i^{inter} | \mu_i^{inter}, diag((\sigma_i^{inter})^2)\right), \quad (5)$$

$$q(\mathbf{z}_i^{intra}) = \mathcal{N}\left(\mathbf{z}_i^{intra} | \mu_i^{intra}, diag((\sigma_i^{intra})^2)\right). \quad (6)$$

**Generative Decoder.** Our generative decoder is given by an inner product between latent variables $\mathbf{Z}$. The output of our decoder is a reconstructed adjacency matrix $\hat{\mathbf{A}}$, which is defined as follows:

$$p(\hat{\mathbf{A}}|\mathbf{Z}) = \prod_{i=1}^N \prod_{j=1}^N p(A_{i,j}|\mathbf{z}_i \mathbf{z}_j), \quad (7)$$

where $p(A_{i,j}|\mathbf{z}_i\mathbf{z}_j) = \sigma(\mathbf{z}_i^\top \mathbf{z}_j)$, and $A_{i,j}$ are the elements of $\hat{\mathbf{A}}$. $\sigma(\cdot)$ is the logistic sigmoid function.

**Edge Weights Prediction as the Pre-training Objective.** We use edge weight reconstruction as the training objective to optimize our pre-trained Bɪ-GAE. Specifically, the pre-training optimizer learns to fit the matrices between the input weighted adjacency matrix $\mathbf{A}_{weight}$ and the reconstructed adjacency matrix $\hat{\mathbf{A}}_{weight}$.

$$\mathcal{L} = \text{MSE}(p(\hat{\mathbf{A}}_{weight}|\mathbf{Z}), \mathbf{A}_{weight})) - \text{KL}(q(\mathbf{Z}) \| p(\mathbf{Z})), \quad (8)$$

The loss function of the bipartite graph pre-training has two parts. The first part is MSE loss which measures how well the pre-training model reconstructs the structure of the bipartite graph. KL works as a regularizer in original VGAE, and $p(\mathbf{Z}) = \mathcal{N}(0, 1)$ is a Gaussian prior.

## 4 EXPERIMENTS

During the graph pre-training, the Bɪ-GAE is optimized by the prediction of edge weights in a self-supervised manner. Subsequently, we utilize the

pre-trained sentence representations of Bɪ-GAE to replace those used in state-of-the-art unsupervised summarization backbones. This allows us to assess the effectiveness of the pre-trained sentence representation in downstream tasks.

## 4.1 Downstream Tasks and Datasets

We evaluate our approach on two summarization datasets: the CNN/DailyMail (Hermann et al., 2015) dataset and the Multi-news (Fabbri et al., 2019) dataset. The CNN/DailyMail comprises articles from CNN and Daily Mail news websites, summarized by their associated highlights. We follow the standard splits and preprocessing steps used in baselines (See et al., 2017; Liu and Lapata, 2019; Zheng and Lapata, 2019; Xu et al., 2020; Liang et al., 2021), and the resulting dataset contains 287,226 articles for training, 13,368 for validation, and 11,490 for the test. The Multi-news is a large-scale multi-document summarization (MDS) dataset and comes from a diverse set of news sources. It contains 44,972 articles for training, 5,622 for validation, and 5,622 for testing. Referring to prior works (Fabbri et al., 2019; Liu et al., 2021), we create sentence discourse graphs for each document and cluster them, with each cluster yielding a summary sentence.

## 4.2 Pre-training Datasets

We construct a bipartite graph with word and sentence nodes, determining edge weights through graph centrality. The centrality-based weights denoted as $\mathbf{A}_{weight}$ serve as inputs for the Bɪ-GAE model. During pre-training, we use MSE loss to measure the average squared difference between the predicted edge values $\hat{\mathbf{A}}_{weight}$ and the true values input $\mathbf{A}_{weight}$, as it indicates more minor errors between the predicted and true values. We conveniently utilize training datasets without their summarization labels as the corpus to pre-train sentence representations by our Bɪ-GAE.

## 4.3 Backbones of Summarization Approaches

There are several simple unsupervised summarization extraction frameworks, including TextRank (Mihalcea and Tarau, 2004) and LexRank (Erkan and Radev, 2004), as well as more robust graph-based ranking methods such as PᴀᴄSᴜᴍ (Zheng and Lapata, 2019), FAR (Liang et al., 2021), DASG (Liu et al., 2021). Graph-based ranking methods take sentence representations as input, using the algorithm of graph-based sentence

centrality ranking for sentence selection. We now introduce extractive summarization backbones.

- TextRank and LexRank utilize PageRank to calculate node centrality based on a Markov chain model recursively.

- PᴀᴄSᴜᴍ (Zheng and Lapata, 2019) constructs graphs with directed edges. The rationale behind this approach is that the centrality of two nodes is influenced by their relative position in the document, as illustrated by Equation 15.

- DASG (Liu et al., 2021) selects sentences for summarization based on the similarities and relative distances among neighbouring sentences. It incorporates a graph edge weighting scheme to Equation 15, using a coefficient that maps a pair of sentence indices to a value calculated by their relative distance.

- FAR (Liang et al., 2021) modifies Equation 15 by applying a facet-aware centrality-based ranking model to filter out insignificant sentences. FAR also incorporates a similarity constraint between candidate summary representation and document representation to ensure the selected sentences are semantically related to the entire text, thereby facilitating summarization.

The main distinction among the extractive frameworks mentioned above lies in their centrality algorithms. A comprehensive comparison of these algorithms can be found in Appendix 8.

## 4.4 Compared Sentence Embeddings

We evaluate three sentence representations for computing sentence centrality. The first compared sentence embedding employs a **TF-IDF-based approach**, where each vector dimension is calculated based on the term frequency (TF) of the word in the sentence and the inverse document frequency (IDF) of the word across the entire corpus of documents. The second representation is based on the **Skip-thought model** (Kiros et al., 2015), an encoder-decoder model trained on surrounding sentences using a sentence-level distributional hypothesis (Kiros et al., 2015). We utilize the publicly available skip-thought model[3] to obtain sentence representations. The third approach relies on **BERT** (Devlin et al., 2019) or **RoBERTa** (Liu et al., 2019) to generate sentence embeddings.

---

[3] https://github.com/ryankiros/skip-thoughts

Table 1: ROUGE F1 performance of the single document extractive summarization on the CNN/DailyMail. [b] is reported in Xu et al. (2020), [†] is reported in Zheng and Lapata (2019) and [‡] is reported in Liang et al. (2021). [*] means our careful re-implementation due to the absence of publicly accessible source code for these methods or the experiment was missing from the published paper. The best results are **in-bold**.

| Method | LM | ROUGE-1 | ROUGE-2 | ROUGE-L |
|---|---|---|---|---|
| ORACLE | | *54.70* | *30.40* | *50.80* |
| LEAD-3 | | 40.49 | 17.66 | 36.75 |
| TextRank | STVec[†] | 31.40 | 10.20 | 28.60 |
| | TF-IDF[†] | 33.20 | 11.80 | 29.60 |
| | BERT[†] | 30.80 | 9.60 | 27.40 |
| | BI-GAE | **36.60** | **14.58** | **32.91** |
| LexRank | STVec[*] | 31.91 | 10.33 | 28.36 |
| | TF-IDF[‡] | 34.68 | 12.82 | 31.12 |
| | BERT[*] | 27.50 | 7.38 | 24.63 |
| | BI-GAE | **39.73** | **16.81** | **36.02** |
| PACSUM | STVec | 38.60 | 16.10 | 35.30 |
| | TF-IDF | 39.20 | 16.30 | 35.30 |
| | BERT | 40.70 | 17.80 | 36.90 |
| | BERT[b] | 40.69 | 17.82 | 36.91 |
| | RoBERTa[b] | 40.74 | 17.82 | 36.96 |
| | BI-GAE | **41.29** | **18.22** | **37.49** |
| FAR | BERT[*] | 40.83 | 17.85 | 36.91 |
| | RoBERTa[*] | 40.87 | 17.42 | 36.31 |
| | BI-GAE | **41.26** | **18.14** | **37.40** |
| DASG | BERT[*] | 40.89 | 17.68 | 37.10 |
| | RoBERTa[*] | 40.90 | 17.76 | 37.12 |
| | BI-GAE | **41.37** | **18.25** | **37.56** |

Table 2: ROUGE F1 performance of the multi-document extractive summarization on the Multi-News. [°] is reported in Fabbri et al. (2019), [†] is reported in Li et al. (2020) and [‡] is reported in Wang et al. (2020). [*] means our careful implementation due to the absence of publicly accessible source code for these methods or the experiment was missing from the published paper. The best results are **in-bold**.

| Method | LM | ROUGE-1 | ROUGE-2 | ROUGE-L |
|---|---|---|---|---|
| ORACLE | | *52.32* | *22.32* | *47.93* |
| FIRST-3 | | 40.21 | 12.13 | 37.13 |
| TextRank | TF-IDF[°] | 38.44 | 13.10 | 13.50 |
| | BERT[‡] | 41.95 | 13.86 | 38.07 |
| | BERT[*] | 42.56 | 13.69 | 38.47 |
| | BI-GAE | **43.20** | **14.76** | **38.95** |
| LexRank | TF-IDF[°] | 38.27 | 12.70 | 13.20 |
| | TF-IDF[†] | 41.01 | 12.69 | 18.00 |
| | BERT[‡] | 41.77 | 13.81 | 37.87 |
| | BERT[*] | 40.97 | 12.93 | 37.21 |
| | BI-GAE | **42.91** | **14.28** | **38.83** |
| PACSUM | BERT | 43.27 | 14.16 | 38.25 |
| | RoBERTa[*] | 41.33 | 13.33 | 37.59 |
| | BI-GAE | **43.53** | **14.42** | **39.26** |
| DASG | BERT | 42.60 | 13.22 | 16.15 |
| | RoBERTa[*] | 41.73 | 13.33 | 37.59 |
| | BI-GAE | **43.39** | **14.27** | **39.22** |
| FAR | BERT[*] | 43.40 | 14.35 | 36.26 |
| | RoBERTa[*] | 43.08 | 14.07 | 39.00 |
| | BI-GAE | **43.58** | **14.58** | **39.30** |

## 4.5 Implementation Details and Metrics

In pre-training the BI-GAE, we choose the best model and hyper-parameters based on their performance on the validation set. Appendix 8.2 provides detailed information on the hyper-parameters used during the BI-GAE pre-training procedure. For fine-tuning the unsupervised extractive summarization frameworks, there are a few hyper-parameters to be tuned for computing centrality scores. The main hyper-parameters for the extractive summarization frameworks are listed in Appendix 8.3. We have kept the remaining hyper-parameters in the backbones of summarization frameworks unchanged.

## 5 Results and Analysis

### 5.1 Single-Document Experiments

Our results on the CNN/Daily Mail are summarized in Table 1. The Oracle upper bound extracts gold standard summaries by greedily selecting sentences that optimize the mean of ROUGE-1 and ROUGE-2 scores. The results indicate the following: (i) Our pre-trained sentence representation, which incorporates TextRank and LexRank extractive frameworks, yields prominent improvements in ROUGE-1/2/L performances. (ii) Our model acquires intra-sentential distinctive and inter-sentential cohensive features of sentences via pre-training on sentence-word bipartite graphs, aiding graph-based ranking for unsupervised summarization. (iii) Our pre-trained sentence representation outperforms all other robust sentence representation methods across all summarization frameworks.

In contrast, sentence representations initialized with BERT or RoBERTa perform poorly in TextRank and LexRank frameworks. This could be attributed to the collapse of BERT-derived sentence representations, resulting in high similarity scores for all sentences and thus failing to leverage the potential centrality in TextRank and LexRank. However, our methods surpass BERT and RoBERTa in

Table 3: ROUGE F1 performance of the extractive summarization. Pre-trained encoder in our Bɪ-GAE is equipped with one kind of GCNs (GCN$^{inter}$ or GCN$^{intra}$). FAR and DASG are two extractive frameworks, respectively, and are tested in the CNN/DailyMail dataset. The pre-training corpora used also is the downstream CNN/DailyMail dataset without summarization labels.

| Method | LM | ROUGE-1 | ROUGE-2 | ROUGE-L |
|---|---|---|---|---|
| Bɪ-GAE | w. GCN$^{inter}$ | 41.18 | 18.18 | 37.37 |
| | w. GCN$^{intra}$ | 41.20 | 18.19 | 37.40 |
| Bɪ-GAE | w. GCN$^{inter}$ | 41.27 | 18.15 | 37.46 |
| | w. GCN$^{intra}$ | 41.26 | 18.13 | 37.44 |

the FAR and DASG summarization frameworks, showcasing the effectiveness of sentence representations pre-trained by our graph auto-encoders.

## 5.2 Multi-Document Experiments

Table 2 shows the comparison of Multi-news summarization. Given that all frameworks employing our pre-trained representations outperform the Fɪʀsᴛ-3 baseline, our approach effectively mitigates position bias (Dong et al., 2021). This bias often results in incomplete summaries that neglect essential information located in the middle of the document. The results demonstrate two key findings: (i) Our method adeptly captures essential summary-worthy sentences, thereby consolidating the process of sentence clustering and, in turn, improving extractive accuracy. (ii) The embedded, intra-sentential distinctive features and inter-sentential cohensive features are crucial in ranking significant sentences across multiple documents.

## 5.3 Component-wise Analysis

To comprehend how modelling intra-sentential features and inter-sentential features contribute to sentence-word bipartite graphs, we conducted an ablation study on the CNN/DailyMail dataset. As shown in Table 3, we can observe that the Bɪ-GAE model equipped solely with GCN$^{inter}$ or solely with GCN$^{intra}$ performs well. When combined with both, Bɪ-GAE yields the best results across all metrics. This highlights the importance of incorporating intra-sentential and inter-sentential features for effective summarization. Combining the two GCNs leads to complementary effects, enhancing the model's overall performance. On the contrary, using only GCN$^{inter}$ or GCN$^{intra}$ individually results in poor performance, as it fails to capture either the semantically cohensive or the distinctive

Table 4: ROUGE F1 performance of Bɪ-GAE on the downstream CNN/DailyMail summarization and Bɪ-GAE is pre-trained on the Multi-news dataset.

| Method | ROUGE-1 | ROUGE-2 | ROUGE-L |
|---|---|---|---|
| TextRank | 36.69 ↑ 0.09 | 14.91 ↑ 0.33 | 33.19 ↑ 0.28 |
| LexRank | 40.13 ↑ 0.40 | 17.16 ↑ 0.35 | 36.41 ↑ 0.39 |
| PᴀᴄSᴜᴍ | 41.12 ↓ 0.17 | 18.09 ↓ 0.13 | 37.34 ↓ 0.15 |
| FAR | 41.17 ↓ 0.03 | 18.18 ↑ 0.01 | 37.41 ↑ 0.02 |
| DASG | 41.27 ↓ 0.10 | 18.16 ↓ 0.07 | 37.50 ↓ 0.06 |

Table 5: ROUGE F1 performance on the downstream Multi-news extractive summarization, in which the model is pre-trained on the CNN/DailyMail dataset.

| Method | ROUGE-1 | ROUGE-2 | ROUGE-L |
|---|---|---|---|
| TextRank | 43.27 ↑ 0.07 | 14.76 ↑ 0.00 | 39.02 ↑ 0.07 |
| LexRank | 42.87 ↓ 0.04 | 14.28 ↑ 0.00 | 38.81 ↓ 0.02 |
| PᴀᴄSᴜᴍ | 43.46 ↓ 0.07 | 14.52 ↑ 0.10 | 39.26 ↑ 0.00 |
| DASG | 43.27 ↓ 0.12 | 14.43 ↑ 0.16 | 39.15 ↓ 0.07 |
| FAR | 43.54 ↓ 0.04 | 14.61 ↑ 0.03 | 39.30 ↑ 0.00 |

content of the document.

## 5.4 Effects of Pre-training Datasets

To evaluate the impact of different pre-training datasets, we test summarization frameworks using two types of representations pre-trained on distinct corpora. In Table 4 and Table 5, we can observe pre-training on the Multi-news dataset showed minimal performance degradation or limited changes in CNN/DailyMail summarization, and vice versa — the similarity between the two news corpora leads to consistent results in downstream tasks.

## 5.5 Density Estimation of Summarization

There are three measures - density, coverage, and compression - introduced by Grusky et al. (2018) and Fabbri et al. (2019) to assess the extractive nature of an extractive summarization dataset. In this paper, we adopt these measures to evaluate the quality of extracted summaries, as illustrated in Figure 3. The coverage (x-axis) measure assesses the degree to which a summary is derived from the original text. The density (y-axis) measures the extent to which a summary can be described as a series of extractions. Compression $c$, on the other hand, refers to the word ratio between two texts - Text A and Text B. Higher compression pose a challenge as it necessitates capturing the essential

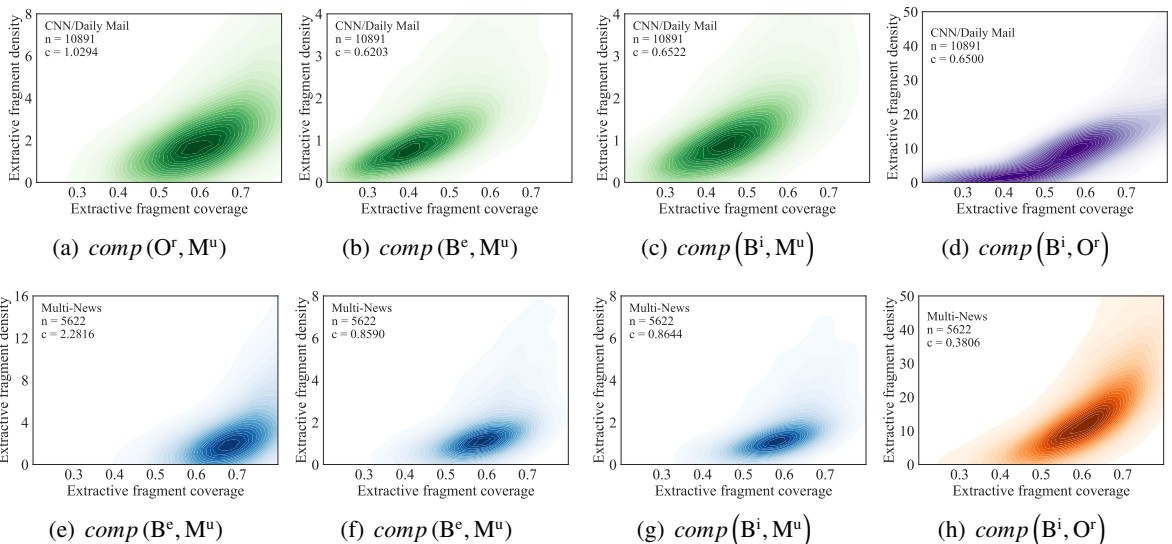

(a) $comp(O^r, M^u)$  (b) $comp(B^e, M^u)$  (c) $comp(B^i, M^u)$  (d) $comp(B^i, O^r)$

(e) $comp(B^e, M^u)$  (f) $comp(B^e, M^u)$  (g) $comp(B^i, M^u)$  (h) $comp(B^i, O^r)$

Figure 3: Density and coverage distributions of extractive compression scores on CNN/DailyMail (subfigures (a), (b), (c), (d)) and Multi-News (subfigures (e), (f), (g), (h)) datasets. Each box represents a normalized bivariate density plot, showing the extractive fragment coverage on the x-axis and density on the y-axis. The top left corner of each plot shows the number $n$ of text and the median compression ratio $c$ between text A and text B. The $comp(A, B)$ denotes the comparison elements are the text A used for comparing, and the text B used as the reference. $comp(O^r, M^u)$: the Oracle and the manual summary. $comp(B^e, M^u)$: the extracted summary of BERT-based DASG and the manual summary. $comp(B^i, M^u)$: the extracted summary of our BI-GAE based DASG and the manual summary. $comp(B^i, O^r)$: the extracted summary of our BI-GAE based DASG and the Oracle.

aspects of the reference text with precision. For detailed mathematical definitions of these evaluation measures, please refer to Appendix 8.6.

We utilize three measures that quantify the level of text overlap between (i) the Oracle summary and the manual summary (subfigures (a) and (e)), (ii) the summary extracted by the BERT-based DASG and the manual summary (subfigure (b) and (f)), (iii) the summary extracted by our BI-GAE based DASG and the manual summary (subfigure (c) and (g)), and (iv) the summary extracted by our BI-GAE based DASG and the Oracle (subfigure (d) and (h)). These measures are plotted using kernel density estimation in Figure 3. Among them, subfigure (a) displays the comparison between the Oracle summary compared to the manual summary, which serves as the upper bound for the density and coverage distributions of extractive compression score in extractive summarization. Subfigure (e) shows this score in the multi-news dataset.

Comparing the extractive summary of our BI-GAE based DASG (DASG integrated by the sentence representation of our BI-GAE) and the extractive Oracle summary in subfigures (a), (b), and (c), we have observed variability in copied word percentages for diverse sentence extraction in CNN/DailyMail. A lower score on the x-axis

suggests a greater inclination of the model to extract fragments (novel words) that differ from standard sentences. Our model also outperforms the BERT-based DASG in compression score (0.6522) to compare subfigures (b) and (c). Regarding the y-axis (fragment density) in subfigure (d), our model shows variability in the average length of copied sequences to the Oracle summary, suggesting varying styles of word sequence arrangement. These advantages persist in the multi-news dataset.

## 6  Conclusion

In this paper, we introduce a pre-training process that optimizes summary-worthy representations for extractive summarization. Our approach employs graph pre-training autoencoders to learn intra-sentential and inter-sentential features on sentence-word bipartite graphs, resulting in pre-trained embeddings useful for extractive summarization. Our model is easily incorporated into existing unsupervised summarization models and outperforms salient BERT-based and RoBERTa-based summarization methods with predominant ROUGE-1/2/L score gains. Future work involves exploring the potential of our pre-trained sentential representations for other unsupervised extractive summarization tasks and text-mining applications.

# 7 Limitations

We emphasize the importance of pre-trained sentence representations in learning meaningful representations for summarization. In our approach, we pre-train the sentence-word bipartite graph by predicting the edge betweenness score in a self-supervised manner. Exploring alternative centrality scores (such as TF-IDF score or current-flow betweenness for edges) as optimization objectives for MSE loss would be a viable option.

Additionally, we seek to validate the effectiveness of the sentence representations learned from BI-GAE in other unsupervised summarization backbones and tasks.

## Acknowledgements

This work is supported by the National Natural Science Foundation of China (No.U20B2053 and No.62376270).

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

# 8 Appendix

## 8.1 Details about Centrality Algorithms

The key idea of graph-based ranking is to calculate the centrality score of each sentence (or vertex) described in section 2.1. In this section, we give the differences in centrality algorithms among several salient summarization backbone models.

The PACSUM (Zheng and Lapata, 2019) method enhances the centrality of two nodes in sentence graphs, considering how their relative positions in a document influence their importance.

$$Centrality(s_i) = \lambda_1 \sum_{j<i} e_{i,j} + \lambda_2 \sum_{j>i} e_{i,j}, \quad (9)$$

where hyper-parameters $\lambda_1$, $\lambda_2$ are different weights for forward and backward-looking directed edges and $\lambda_1 + \lambda_2 = 1$. $e_{i,j}$ is the normalized similarity score.

The **FAR** (Liang et al., 2021) approach enhances the centrality of two nodes with distance constraints in sentence graphs by considering how their relative positions in a document influence their importance.

$$Centrality(s_i) = \lambda_1 \sum_{j<i} Max((e_{i,j} - \epsilon), 0) \\ + \lambda_2 \sum_{j>i} Max((e_{i,j} - \epsilon), 0), \quad (10)$$

where $\epsilon = \beta \cdot \left(max(e_{i,j}) - min(e_{i,j})\right)$. For $s_1$, the threshold $\epsilon$ can be seen as a diameter, $s_1$ is the centre. $\beta$ is a hyper-parameter to control the scale of diameter.

The **DASG** (Liu et al., 2021) method enhances the centrality of two nodes in sentence graphs by taking into account their relative position and semantic facets within a document.

$$Centrality(s_i) = \lambda^+_{\left\lfloor \frac{j-i}{m}+1 \right\rfloor} \sum_{j<i} e_{i,j} \\ + \lambda^-_{\left\lfloor \frac{i-j}{m}+1 \right\rfloor} \sum_{j>i} e_{i,j}, \quad (11)$$

where $\lambda^+_1, ..., \lambda^+_k$ and $\lambda^-_1, ..., \lambda^-_k$ are fixed hyper-parameters and $k$ is set to be 3 empirically.

## 8.2 Hyper-parameters in BI-GAE pre-training

We mainly use PyTorch Geometric, PYG [4] to implement BI-GAE. More specifically, we limit the

vocabulary to 50,000 and initialize tokens with 300-dimensional GloVe 840B embeddings[5]. We filter stop words and punctuations when creating word nodes and truncate the input document to a maximum length of 50 sentences. To eliminate the noisy common words, we remove 10% of the vocabulary with low TF-IDF values over the whole dataset. We initialize sentence nodes with $d_s = 150$. We use a batch size of 8 during pre-training and apply the Adam optimizer with a learning rate of 5e-5 for CNN/DailyMail and 2e-5 for Multi-News. The dropout is 0.1. The pre-training model is trained for 210,000 steps, and the warm-up step is set to 8000. Attempts made to invoke certain model interfaces in PYG have revealed that using JKNET (Xu et al., 2018) and GCNII (Chen et al., 2020) as the encoder backbone in the pre-training process results in performance for downstream tasks that are essentially indistinguishable from those of GCN.

## 8.3 Hyper-parameters in Summarization

We begin by using Stanford NLP [6] to split sentences and preprocess the dataset. The source text has a maximum sentence length of 512, while the summary is limited to a maximum sentence length of 140. During the tuning process for extractive summarization, we fine-tune the parameters related to the centrality algorithm within a narrow range of [-1.0, 2.0]. Table 6 presents the optimal hyper-parameters for each extractive summarization backbones, utilizing our BI-GAE pre-trained sentence representations. For the CNN/DailyMail dataset, we select the top-3 sentences for the summarization based on the average length of the Oracle human-written summaries, whereas, for Multi-New, we choose the top-9 sentences.

## 8.4 Sentence Similarity Computation

The crucial aspect of the unsupervised graph rank method in downstream tasks lies in the calculation of similarity between two sentences. In this regard, we examine two methods for calculating similarity, both of which draw inspiration from the similarity calculation approach utilized in PACSUM(Zheng and Lapata, 2019). The first one can employ a pair-wise dot product to compute an unnormalized similarity matrix $\bar{E}_{ij} = v_i^\top v_j$, and the second one is cosine similarity $\bar{E}_{ij} = cos(v_i, v_j)$. The final normalized

---

[4] https://github.com/pyg-team/pytorch_geometric

[5] https://nlp.stanford.edu/projects/glove/
[6] https://github.com/stanfordnlp/CoreNLP

| Datasets | Methods | Hyper-parameters |
|---|---|---|
| CNN/DailyMail | PacSum | $\lambda_1 = -1.0, \lambda_2 = 1.0$ |
| | FAR | $\lambda_1 = -0.5, \lambda_2 = 0.9$ |
| | DASG | $\beta = 0.05, \lambda_1^+ = -1.5, \lambda_2^+ = -0.5, \lambda_3^+ = -1.0, \lambda_1^- = 1.0, \lambda_2^- = 1.5, \lambda_3^- = 2.0$ |
| Multi-News | PacSum | $\lambda_1 = 0.3, \lambda_2 = -0.7$ |
| | FAR | $\lambda_1 = -0.5, \lambda_2 = 2.0$ |
| | DASG | $\beta = 0.8, \lambda_1^+ = -1.5, \lambda_2^+ = -0.5, \lambda_3^+ = -1.0, \lambda_1^- = 1.0, \lambda_2^- = 1.5, \lambda_3^- = 2.0$ |

Table 6: Main hyper-parameters of centrality algorithms for tuning extractive summarization with our Bɪ-GAE pre-trained sentence representations.

Table 7: ROUGE F1 performance of the extractive summarization. The pre-trained encoder in our Bɪ-GAE is equipped with extractive frameworks DASG or FAR, respectively, and is tested in CNN/DailyMail dataset. The pre-training corpora used also is CNN/DailyMail dataset without summarization labels.

| Method | Sim | ROUGE-1 | ROUGE-2 | ROUGE-L |
|---|---|---|---|---|
| Bɪ-GAE + DASG | cos | 41.13 | 17.97 | 37.34 |
| | dot | 41.37 | 18.25 | 37.56 |
| Bɪ-GAE + FAR | cos | 41.20 | 18.19 | 37.40 |
| | dot | 41.26 | 18.24 | 37.45 |

Table 8: ROUGE F1 performance of the extractive summarization. The pre-trained encoder in our Bɪ-GAE is equipped with extractive frameworks DASG or FAR, respectively, and is tested in the Multi-news dataset. The pre-training corpora used is the Multi-news dataset without summarization labels.

| Method | Sim | ROUGE-1 | ROUGE-2 | ROUGE-L |
|---|---|---|---|---|
| Bɪ-GAE + DASG | cos | 43.39 | 14.27 | 39.22 |
| | dot | 43.12 | 14.16 | 38.99 |
| Bɪ-GAE + FAR | cos | 42.97 | 14.34 | 38.87 |
| | dot | 43.58 | 14.58 | 39.30 |

similarity matrix E is defined as:

$$\tilde{E}_{ij} = \bar{E}_{ij} - \left[ min\bar{E} + \beta(max\bar{E} - min\bar{E}) \right], \quad (12)$$

where $\tilde{E}_{ij}$ is designed to mitigate the influence of absolute values and instead emphasize the relative contributions of different similarity scores. The hyper-parameter $\beta \in [0, 1]$ controls the threshold below which the similarity score of $\tilde{E}_{ij}$ is set to 0.

Figure 7 and Figure 8 illustrate the testing results of models using two similarities. Through empirical analysis, we have discovered that the pair-wise dot product yields better performance in most cases on CNN/Dailymail summarization and Multi-news summarization. This finding aligns with the results reported in PᴀᴄSᴜᴍ(Zheng and Lapata, 2019).

## 8.5 Bɪ-GAE Pre-training Validation

We meticulously fine-tune a multitude of parameters in our process. For the pre-training of the CNN/Daily Mail corpus, we find that the optimal learning rate for our model is 5e-5, with a batch size of 8. Similarly, in the pre-training of the Multi-news corpus, we find that the optimal learning rate is 2e-5 while maintaining a batch size of 8.

To assess the pre-training performances, we conduct accuracy tests of the edge weight prediction on the verification set. As shown in Figure 4, our findings indicate that the optimal prediction accuracy for both corpora typically ranges between 0.60 and 0.65. Based on these observations, we formulated the following hypothesis: when there are more unique nodes, their edge weights should be smaller since they are not shared by other nodes. Conversely, when there are more shared nodes, their edge weights should be greater. The improvement in performance on downstream tasks validates the soundness of our hypothesis.

## 8.6 Characterizing Summarization Strategies

As shown in Figure 3, each box is a normalized bivariate density plot of extractive fragment coverage (x-axis) and density (y-axis), and the top left corner of each plot shows the median compression ratio $c$ between text A and text B.

**Fragment Coverage** Extractive fragment coverage is the percentage of words in the summary that are from the source article, measuring the extent to which a summary is derivative of a text:

$$COVERAGE(A, B) = \frac{1}{|B|} \sum_{f \in F(A,B)} |f|, \quad (13)$$

where $F(A, B)$ is the set of shared sequences of

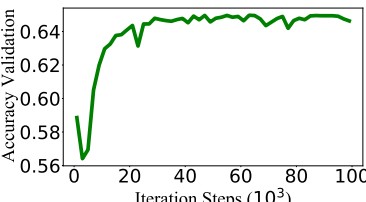

(a) Pre-training on CNN/Daily Mail

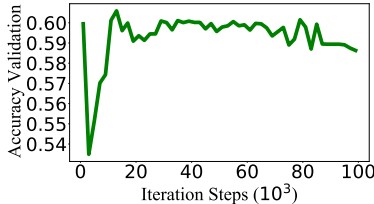

(b) Pre-training on Multi-news

Figure 4: Verification results of edge prediction accuracy during BI-GAE pre-training on CNN/Daily Mail and Multinews corpora.

text and article text to include: the comparison between extracted summary and manual summary, the comparison between the extractive Oracle and the manual summary, or the comparison between extracted summary and Oracle summary.

tokens in A and B and is identified as extractive in a greedy manner. For example, a summary (text B) with 10 words that 7 words are the same as its article (text A) and include 3 new words will have $COVERAGE(A, B)$ =0.7.

**Fragment Density** The density measure quantifies the average length of the extractive fragment to which each word in the text belongs.

$$\mathcal{DENSITY}(A, B) = \frac{1}{|B|} \sum_{f \in F(A,B)} |f|^2 . \quad (14)$$

For instance, a summary (text B) might contain many individual words from the article (text A) and therefore have high coverage. For instance, a summary might contain many individual words from the article and therefore have high coverage. For an article (text A) with a 10-word summary (text B) made of two extractive fragments of lengths 3 and 4 would have COVERAGE(A, S) = 0.7 and $\mathcal{DENSITY}(A, B)$ =2.5.

**Compression Ratio** The compression ratio $c$ is defined as the word ratio between the article and summary:

$$COMPRESSION(A, B) = \frac{|A|}{|B|}. \quad (15)$$

Summarizing with higher compression is challenging as it requires capturing more precisely the critical aspects of the article text.

Among our settings about the above metrics, we have expanded the comparison between summary