# OpenReview forum: "Bipartite Graph Pre-training for Unsupervised Extractive Summarization with Graph Convolutional Auto-Encoders"
_EMNLP/2023/Conference — EMNLP 2023 Findings_

### Official Review · Reviewer_vmKw · 2023-07-31

**Soundness:** 4

**Excitement:**

4: Strong: This paper deepens the understanding of some phenomenon or lowers the barriers to an existing research direction.

**Missing References:**

Sun, Shichao, et al. "Improving Sentence Similarity Estimation for Unsupervised Extractive Summarization." *ICASSP 2023-2023 IEEE International Conference on Acoustics, Speech and Signal Processing (ICASSP)*. IEEE, 2023.

@inproceedings{sun2023improving,
  title={Improving Sentence Similarity Estimation for Unsupervised Extractive Summarization},
  author={Sun, Shichao and Yuan, Ruifeng and Li, Wenjie and Li, Sujian},
  booktitle={ICASSP 2023-2023 IEEE International Conference on Acoustics, Speech and Signal Processing (ICASSP)},
  pages={1--5},
  year={2023},
  organization={IEEE}
}

**Paper Topic And Main Contributions:**

The paper introduces a pioneering approach to pre-train sentence embeddings using a graph-based auto-encoder. The proposed method effectively bridges the gap between pre-training and sentence-ranking tasks in unsupervised summarization. By leveraging intra-sentential and inter-sentential features from a sentence-word bipartite graph, the pre-trained embeddings exhibit promising utility in extractive summarization. Notably, the experimental results demonstrate that these embeddings can be seamlessly integrated into various sentence-ranking methods, such as TextRank, PacSum, and FAR, yielding superior performance compared to traditional embeddings like BERT and RoBERTa.

**Reasons To Accept:**

1. **In-Depth Exploration of Graph Pre-training Auto-Encoder:** The paper presents a thorough investigation into the graph-based pre-training auto-encoder, enabling the learning of both intra-sentential and inter-sentential features. This unique approach contributes to the development of highly valuable pre-trained embeddings specifically tailored for extractive summarization.
2. **Versatility and Effectiveness in Different Summarization Methods:** By showcasing the adaptability of the proposed embeddings in diverse unsupervised summarization techniques, the paper substantiates the practicality of its method. The experimental results provide strong evidence of the embeddings' effectiveness in two extractive summarization datasets, further strengthening its credibility.

**Reasons To Reject:**

1. **Confusing Textual Description:** Some portions of the paper lack clarity and context. For instance, the text "These fine-tuned sentence embeddings" in line 17 is ambiguous and does not provide sufficient explanation regarding the fine-tuning process for sentence embeddings. Enhancing the clarity of such descriptions will ensure a better understanding of the methodology.
2. **Limited Experimentation Domains:** While the paper exhibits promising results on the news domain dataset, it would be beneficial to broaden the scope of experimentation to include various domains, such as scientific articles (arxiv summarization dataset) and dialogue summarization (ICSI dataset or AMI dataset). Conducting experiments in diverse domains will enhance the generalizability and robustness of the proposed method.

**Reproducibility:**

4: Could mostly reproduce the results, but there may be some variation because of sample variance or minor variations in their interpretation of the protocol or method.

**Reviewer Confidence:**

3: Pretty sure, but there's a chance I missed something. Although I have a good feel for this area in general, I did not carefully check the paper's details, e.g., the math, experimental design, or novelty.

---

> ### Author Rebuttal · Authors · 2023-08-27
>
> Reviewer vmKw:
>
> We would like to thank you for strongly supporting our paper! [Reviewer psgw, VXBy] have also affirmed our contributions, such as:
> 1. An interesting approach [Reviewer psgw]
> 2. Provides a detailed explanation [Reviewer psgw]
> 3. Making it easier to understand and replicate the experiments [Reviewer psgw]
> 4. Clear and concise description [Reviewer psgw] Well-written paper, easy to follow [Reviewer VXBy]
> 5. Experiment setup and comparison is clear [Reviewer VXBy]
> 6. A new self-supervised learning method for extractive text summarization [Reviewer VXBy]
>
> Here we will carefully address your concerns in detail.
>
> - The dataset we chose is an intersection of the datasets used in several baselines in the original papers. These baselines are evaluated on both single-document summarization and multi-document summarization tasks.
>
> - As far as I know, dialogue summarization datasets (ICSI or AMI dataset) are aligned with abstractive summarization, making them more suitable for validating abstractive summarization methods. Furthermore, our baselines have yet to be experimented with or have their performance results disclosed on these two datasets. Reproducing the results of baselines on a new dataset would entail costs and meticulous replication.
>
> - Besides, the ARXIV dataset has domain disparities compared to news corpora. Whether the pretraining on news-based corpora is also warranted in a new domain becomes a topic worthy of exploration. For experiments on new-domain datasets, the initial step involves constructing a bipartite graph based on the source documents, followed by bipartite graph pretraining and downstream summarizaztion. Considering the invisible workload, more complementary experiments（like other extractive summarization datasets）can be explored in future endeavours.
>
> - Our pretraining approach was conducted on news-related datasets (the downstream news summarization dataset we employed). The BERT and RoBERTa models we are comparing were also primarily pre-trained on extensive news corpora. This ensures a relatively fair domain consistency for our pretraining corpus.
>
> - Missing References will be cited in our final manuscript.
>
> ---
>
> Thank you for all the constructive feedback and positive comments.
>
> Sincerely & Best wishes

---

### Official Review · Reviewer_VXBy · 2023-08-04

**Soundness:** 3

**Excitement:**

3: Ambivalent: It has merits (e.g., it reports state-of-the-art results, the idea is nice), but there are key weaknesses (e.g., it describes incremental work), and it can significantly benefit from another round of revision. However, I won't object to accepting it if my co-reviewers champion it.

**Missing References:**

Zhao, Jinming, et al.  "Summpip: Unsupervised multi-document summarization with sentence graph compression." In Proceedings of the 43rd international acm sigir conference on research and development in information retrieval, pp. 1949-1952. 2020.

**Paper Topic And Main Contributions:**

This paper proposed a graph auto-encoder based pre-training method to obtain sentence representation, the graph is constructed by considering both intra and inter sentence word connection. Compared with other representation methods such as TFIDF and BERT, the proposed method works better when fed with graph ranking models for extractive text summarisation.

**Questions For The Authors:**

Question A: What is the computation cost for building the word sentence graph?  what could be the maximum input length when using your method?

**Reasons To Accept:**

* Well written paper, easy to follow.
* Experiment setup and comparison is clear.
* A new self-supervised learning method for extractive text summarization.

**Reasons To Reject:**

* Since BERT and Roberta is pre-trained with local context, it's better to compare with other "summarization-focused" pre-trained method, for example, take the encoder part of BART or PEGASUS.
* Apart from ROUGE, more evaluation metrics for summary redundancy/coherence is needed. Also consider some human evaluation.
* The method is only evaluated with ranking based models, not sure whether it works for clustering based approach such as [1].

[1] Zhao, Jinming, et al. "Summpip: Unsupervised multi-document summarization with sentence graph compression." In Proceedings of the 43rd international acm sigir conference on research and development in information retrieval, pp. 1949-1952. 2020.


**Reproducibility:**

4: Could mostly reproduce the results, but there may be some variation because of sample variance or minor variations in their interpretation of the protocol or method.

**Reviewer Confidence:**

4: Quite sure. I tried to check the important points carefully. It's unlikely, though conceivable, that I missed something that should affect my ratings.

---

> ### Author Rebuttal · Authors · 2023-08-27
>
> Dear Reviewer VXBy:
>
> We thank you for all the constructive feedback and positive comments. In the following, we address each concern raised by the reviewer individually.
>
> *1. Concern about other 'pre-trained methods'.*
>
> - Thank you for your suggestion！The two models suggested models, BART (Lewis et al., ACL2020) and PEGASUS (Zhang et al., ICML2020), are pre-training methods based on sequence-to-sequence architectures, primarily suitable for supervised abstractive summarization tasks. Whether they can be directly adapted for unsupervised extractive summarization tasks requires further discussion and dedicated design.
>
> *2. Concern about 'more evaluation metrics'.*
>
> - Thanks for your constructive comments. Except for the ROUGE, we also introduce three other metrics: Fragment Coverage, Fragment Density, and Compression Ratio. These three types of metrics have been widely employed as primary indicators to reflect the nature of a summary (by Fabbri et al., ACL2019, Sharma et al., ACL2019, Huang et al., NAACL 2021, Salemi et al., EMNLP 2021). Given this, we have opted for widely acknowledged metrics as the methods for assessing the characteristics of summaries, apart from ROUGE.
>
> - We created a pre-training method to capture informative aspects of sentence representations. As the described motivation of our paper [in Lines 070-074], the existing graph approach (like GCNs by Bruna et al., 2014) emphasizes encoding distinct sentences rather than always informative ones, potentially limiting the extraction of crucial sentences. For a deeper check on whether our model captures informative content, we prefer using Fragment Coverage, Fragment Density, and Compression Ratio to validate summary qualities thoroughly.
>
> - (can be discussed) We concur with the reviewers that coherence is a practical evaluative approach for further analyzing summary characteristics. However, the above three metrics we employ are more aligned with the original intention of our paper's focus on informative summaries than coherence. Moreover, achieving an objective and broadly accepted assessment might prove challenging due to the potential bias and subjectivity of human evaluations. We expect our approach to be validated using human validation and coherence metrics in the future.
>
> *3. Concern about 'not sure whether it works for clustering methods'.*
>
> -   Thank you for your careful review. All summarization backbones introduced in Chapter 9.1 are all sentence-ranking methods. In order to achieve fair comparison and maintain consistency, we have not used graph clustering modules for backbones.
>
> - Our pre-training is customized for summarization to address the "gap" between upstream pre-training and sentence ranking in summarization. In upstream pre-training, we explicitly capture intra-sentential distinct and inter-sentential informative features via sentence-word bipartite graphs. These embeddings aid in identifying distinct and informative sentences during ranking sentences in downstream summarization [our motivation is described in Lines 100-113]. Hence, we emphasize ranking-based models based on our graph pre-training approach. In future work, we can talk about more modules (like graph clustering), but not the focus of our work now.
>
> - Missing references will be cited in the final paper. Thanks again for your appreciation and constructive comments on our work.
>
> *Q\&A.*
>
> Thank you for your question. The computation cost of building the sentence-word bipartite graph is negligible (O(n)). Moreover, its advantages are evident since constructing a sentence-word bipartite graph does not require reliance on external tools or consideration of the error propagation issue.
>
> ---
>
> Thanks again for your appreciation and constructive comments on our work!
>
> Best regards
>
> ---
> **References:**
>
> [1] Mike Lewis et al., BART: Denoising Sequence-to-Sequence Pre-training for Natural Language Generation, Translation, and Comprehension. ACL.2020
>
> [2] Jingqing Zhang et al., PEGASUS: Pre-training with Extracted Gap-sentences for Abstractive Summarization. ICML.2020
>
> [3] Alexander R. Fabbri et al., Multi-News: A Large-Scale Multi-Document Summarization Dataset and Abstractive Hierarchical Model. ACL.2019
>
> [4] Eva Sharma et al., BIGPATENT: A Large-Scale Dataset for Abstractive and Coherent Summarization. ACL.2019
>
> [5] Luyang Huang et al., Efficient Attentions for Long Document Summarization. NAACL-HLT.2021
>
> [6] Alireza Salemi et al., ARMAN: Pre-training with Semantically Selecting and Reordering of Sentences for Persian Abstractive Summarization. EMNLP.2021

---

### Official Review · Reviewer_psgw · 2023-08-07

**Soundness:** 3

**Excitement:**

3: Ambivalent: It has merits (e.g., it reports state-of-the-art results, the idea is nice), but there are key weaknesses (e.g., it describes incremental work), and it can significantly benefit from another round of revision. However, I won't object to accepting it if my co-reviewers champion it.

**Paper Topic And Main Contributions:**

The paper addresses the task of unsupervised extractive summarization using graph-based ranking with pre-trained embeddings. However, the paper lacks novelty as the topic of extractive summarization with graph ranking has been explored extensively in the past. The proposed method does not demonstrate significant advancements over previous approaches, which weakens its contribution to the field.

**Reasons To Accept:**

One strength of the paper is the clear and concise description of the proposed method. The use of graph pre-training auto-encoders to model intra-sentential and inter-sentential features through sentence-word bipartite graphs is an interesting approach. The paper also provides a detailed explanation of how pre-trained sentence embeddings are used in the graph-based ranking algorithm, making it easier to understand and replicate the experiments.

**Reasons To Reject:**

The main weakness of the paper lies in its limited novelty. The topic of extractive summarization using graph ranking is not new, and the proposed method does not demonstrate significant improvements over previous techniques. While the paper claims to outperform BERT- and RoBERTa-based methods, the extent of this improvement is not convincingly demonstrated through rigorous experiments and comprehensive comparisons with state-of-the-art approaches.

**Reproducibility:**

4: Could mostly reproduce the results, but there may be some variation because of sample variance or minor variations in their interpretation of the protocol or method.

**Reviewer Confidence:**

4: Quite sure. I tried to check the important points carefully. It's unlikely, though conceivable, that I missed something that should affect my ratings.

---

> ### Author Rebuttal · Authors · 2023-08-27
>
> Reviewer psgw:
>
> Many thanks for your positive feedback! We are pleased you found our paper: a clear and concise description, an interesting approach, and a detailed explanation. Here, we will carefully address your concerns and questions in detail:
>
> *1. Concern about 'our paper lacks novelty and the topic of extractive summarization using graph ranking is not new'.*
>
> - To the best of our knowledge, **our approach stands as the first effort in introducing the bipartite word-sentence graph pre-training method. Furthermore, we are also the first to employ bipartite graph pre-trained sentence representations in an unsupervised extractive summarization**.
>
> - In fact, **the other two reviewers confirm the novelty of our paper**: "A new self-supervised learning method for extractive text summarization." [Reviewer VXBy] and an "In-Depth Exploration of Graph Pre-training Auto-Encoder" [Reviewer vmKw]. [Reviewer vmKw] also aptly affirmed, "Our paper introduces a pioneering approach to pre-train sentence embeddings using a graph-based auto-encoder."
>
> - Additionally, we would like to reaffirm our initial motivation to clear up any possible confusion. Our intention is not to propose a new unsupervised summarization ranking method but rather to introduce a sentence representation approach suitable for unsupervised summarization, stemming from an upstream pre-training task. In other words, **our improvement is focused on the upstream of unsupervised summarization tasks**.
>
> *2. Concern about 'the improvement is not convincingly demonstrated.'*
>
> - We clarify that Sec 5, Tables 1 and 2 display the superior performance of our approach. Such improvement excludes the influence of irrelevant factors like data fluctuation and hypermeters since the unsupervised paradigm of downstream tasks. We will also release the source code.
>
> - In this context, we have summarized the performance of our model surpassing RoBERTa-based sentence representations in downstream tasks, as shown in the table. Notably, our model demonstrates significant enhancement, with an **average increase of 0.57 and 1.17 points in ROUGE scores across the two datasets**.
>
> CNN/Daily | ROUGE-1 | ROUGE-2 | ROUGE-L
> ---|---|---|---
> ROBERTa (+PacSum) | 40.74 | 17.82 | 36.96 |
> Ours (+PacSum) | 41.29($\uparrow$0.55) | 18.22($\uparrow$0.40) | 37.49($\uparrow$0.53) |
> ROBERTa (+FAR) | 40.87 | 17.42 | 36.31 |
> Ours (+FAR)| 41.26($\uparrow$0.39) | 18.14($\uparrow$0.72) | 37.40($\uparrow$1.09) |
> ROBERTa (+DASG) | 40.90 | 17.76 | 37.12 |
> Ours (+DASG)| 41.37($\uparrow$0.47) | 18.25($\uparrow$0.49) | 37.56($\uparrow$0.44) |
> average gains | **0.47** | **0.54** | **0.69** |
>
>
>
> MultiNews | ROUGE-1 | ROUGE-2 | ROUGE-L
> ---|---|---|---
> ROBERTa (+PacSum) | 41.33 | 13.33 | 37.59 |
> Ours (+PacSum) | 43.53($\uparrow$2.20) | 14.42($\uparrow$1.09)| 39.26($\uparrow$1.67) |
> ROBERTa (+FAR) | 43.08 | 14.07 | 39.00 |
> Ours (+FAR)| 43.58($\uparrow$0.50) | 14.58($\uparrow$0.51) | 39.30($\uparrow$0.30) |
> ROBERTa (+DASG) | 41.73 | 13.33 | 37.59 |
> Ours (+DASG)| 43.39($\uparrow$1.66) | 14.27($\uparrow$0.94) | 39.22($\uparrow$1.63) |
> average gains | **1.45** | **0.85** | **1.20** |
>
>
> - Our pre-training is tailor-made for the summarization task to alleviate the "gap" between upstream pre-training and downstream sentence ranking tasks. However, this gap persists in baseline pre-training methods, mostly unaddressed for unsupervised tasks due to infeasible fine-tuning in unsupervised scenarios. BERT's sentence bias leads to the collapse problem, confirmed by [Yan et al., ACL2020].
>
> - Reviewer 3 has also affirmed our convincing contributions, such as: 'The experimental results prove the embeddings' effectiveness in two extractive summarization datasets, further strengthening its credibility. '
>
> - While our method has achieved exceptional performance, the article does not explicitly claim statistical significance for the improvements. We appreciate your insightful suggestions and plan to conduct rigorous significance testing to explore whether our results truly represent a significant enhancement in the final version of the paper.
>
> ---
> Again, we sincerely appreciate your review and feedback. We hope our responses have provided you with a clearer picture of our work and have addressed your concerns. Besides, we consider the comment 'improvement is not convincingly demonstrated' to be quite broad; we cannot provide any explanations without explicit mentions. We hope to have more explicit opinions and suggestions.
>
> Sincerely & Best wishes
>
> ---
> **References:**
> [1] Yuanmeng Yan, Rumei Li, Sirui Wang, Fuzheng Zhang, Wei Wu, Weiran Xu: ConSERT: A Contrastive Framework for Self-Supervised Sentence Representation Transfer. ACL/IJCNLP (1) 2021: 5065-5075

---

### Meta-Review · Area_Chair_NcNt · 2023-09-25

**Recommendation:** 3

**Metareview:**

The paper is the clear and concise description of the proposed method, However, there are some insufficient works, such as evaluation metrics for summary redundancy/coherence,  human evaluation, etc.
Moreover, more advanced datasets and flexible tasks are needed to contribute the development of extractive summarization, other than CNN/DM or MDS.
Therefore, I would suggest the paper accepted as Findings in the conference.

---

### Decision · Program_Chairs · 2023-10-07

**Decision:**

Accept-Findings

**Comment:**

The paper is the clear and concise description of the proposed method, However, there are some insufficient works, such as evaluation metrics for summary redundancy/coherence,  human evaluation, etc.
Moreover, more advanced datasets and flexible tasks are needed to contribute the development of extractive summarization, other than CNN/DM or MDS.
Therefore, I would suggest the paper accepted as Findings in the conference.